# Prospects of future MeV telescopes
# in probingweak-scale dark matter

Marco Cirelli⋆ and Arpan Kar†

Laboratoire de Physique Théorique et Hautes Énergies (LPTHE),
CNRS & Sorbonne Université, 4 Place Jussieu, Paris, France

⋆ marco.cirelli@gmail.com , † arpankarphys@gmail.com

## Abstract

Galactic weak-scale Dark Matter (DM) particles annihilating into lepton-rich channels not only produce gamma-rays via prompt radiation but also generate abundant energetic electrons and positrons, which subsequently emit through bremsstrahlung or inverse Compton scattering (collectively called "secondary-radiation photons"). While the prompt gamma-rays concentrate at high-energy, the secondary emission falls in the MeV range, which a number of upcoming experiments (AMEGO, e-ASTROGAM, MAST...) will probe. We investigate the sensitivity of these future telescopes for weak-scale DM, focusing for definiteness on observations of the galactic center. We find that they have the potential of probing a wide region of the DM parameter space which is currently unconstrained. Namely, in rather optimistic configurations, future MeV telescopes could probe thermally-produced DM with a mass up to the TeV range, or GeV DM with an annihilation cross section 2 to 3 orders of magnitude smaller than the current bounds, precisely thanks to the significant leverage provided by their sensitivity to secondary emissions. We comment on astrophysical and methodological uncertainties, and compare with the reach of high-energy gamma ray experiments.

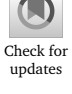

# 1 Introduction

Despite decades of investigation, Dark Matter (DM) remains one of the most pressing unresolved issues in modern cosmology and particle physics [1]. Weak-scale particle Dark Matter (DM), which is here broadly defined as having a mass between a few GeV and tens of TeV's, remains a motivated framework for the solution of the DM problem. This used to be inspired mostly by the fact that several New Physics theories predicted a weak-scale particle with the appropriate properties to play the role of DM. Nowadays, as the promised New Physics has not showed up (at least yet), the motivation mostly shifted to the fact that particles with weak-scale mass and interactions are produced in the right amount in the early Universe. Weak-scale DM is still therefore the subject of intense searches with many different methods [1].

Indirect Detection (ID) techniques aim at detecting the signals of DM particle annihilations or decays in the Galaxy or in different astrophysical environments. For weak-scale DM, and restricting to the signals in photons, the observables can consist of mainly two things. On one side, high-energy $\gamma$-rays are produced directly in the annihilation (or decay) process, hence called *prompt emission*. On the other side, lower-energy X-rays/$\gamma$-rays are produced by the electrons and positrons (generated by DM annihilations or decay), in particular via Inverse Compton Scattering (ICS) processes on the ambient light and via bremsstrahlung processes on the galactic gas. These are dubbed *secondary emission*. In most cases, prompt and secondary emissions both exist, and their relative importance depends on the details of the DM model (for instance, lepton-rich annihilation channels will produce abundant $e^{\pm}$ and thus a significant ICS signal), on the ambient conditions (for instance, regions of the Galaxy where the ambient light is more intense will be a more favorable target to search for ICS photons) as well as on the sensitivity of the dedicated experiments.

In the past, the prompt emission signal from weak-scale DM received by far most of the attention, although the importance of secondary emission in DM indirect detection searches has been highlighted in a number of studies, see e.g. [2–12] for ICS and [13] for bremsstrahlung. One potential difficulty for detecting secondary emission is the relatively poor sensitivity of past and existing telescopes in the range 0.1-100 MeV, the so-called *MeV gap*. Indeed, secondary

photons from weak-scale DM will in general fall in the gap.[1] Fortunately, a number of up-coming or planned MeV telescopes aim at filling this gap [14]. Among them, AMEGO [15–17], e-ASTROGAM [18,19] and MAST [20] are expected to provide a good sensitivity. Our main goal in this work is to explore the potential that these future MeV telescopes have in constraining weak-scale DM, compared in particular to high-energy $\gamma$-ray telescopes.

Recently, a few studies have done work related to this paper. Refs. [21–24] have applied the same principle to another range of DM masses, namely sub-GeV DM. In [25–28] the authors assess the sensitivity of several future MeV telescopes to MeV-GeV DM, including prompt emission only, and [29,30] consider the case of Primordial Black Hole (PBH) dark matter. In [31] the authors consider secondary emission signals for a specific weak-scale DM model and for a specific target (dwarf galaxies). Our work therefore significantly extends and generalizes these studies.

The remainder of the paper is organized as follows. In section 2 we discuss the MeV-GeV photon fluxes, focusing in particular on the inner galactic region. Section 2.1 provides the discussion on the weak-scale DM-induced photon signals in the MeV-GeV energy range, focusing on the prompt (section 2.1.1) and the secondary (sections 2.1.2 and 2.1.3) emissions. Section 2.2 summarizes different photon backgrounds and data in the MeV-GeV range. In section 3 we briefly review the different near-future space-based MeV telescopes considered in this work. Section 4 details the methodology used to derive the bounds and the projected sensitivities of the MeV telescopes on DM annihilation. We then present and discuss the results of our analysis in section 5; we also discuss the effects of propagation of $e^{\pm}$ in the Galaxy. Our conclusions are contained in section 6. Finally, in Appendix A we provide a discussion on the impact of varying different astrophysical ingredients.

## 2 MeV-GeV photons in the Galaxy

The full energy range of the observed photons that we consider here is $0.1\,\mathrm{MeV} \lesssim E_{\gamma} \lesssim 100\,\mathrm{GeV}$, and, in particular, for the future MeV telescopes we use the range $0.2\,\mathrm{MeV} \lesssim E_{\gamma} \lesssim 5\,\mathrm{GeV}$ (for which the galactic background models are provided in [27]). As for the target of observation, we consider a disk of $10°$ radius around the galactic Center (GC) (following [27, 32, 33]), which becomes our region of interest (ROI). Such an angular size of observation is of the same order as the maximum angular width of the upcoming MeV $\gamma$-ray telescopes [15, 16]. Below we discuss the possible DM signals as well as the background photons and observed data from this region in our photon energy range of interest.

### 2.1 Photon signals from DM annihilation in the Galaxy

In this work we assume that the origin of the signal is due to the pair-annihilations of a single-component self-conjugate weak-scale DM, that explains the entire observed DM density. We consider the following annihilation channels one at a time, assuming a 100% branching fraction each time: $\mathrm{DM\,DM} \to \mu^{+}\mu^{-}$, $\mathrm{DM\,DM} \to e^{+}e^{-}$, $\mathrm{DM\,DM} \to b\bar{b}$ and $\mathrm{DM\,DM} \to W^{+}W^{-}$.

We assume that the density distribution of DM in the halo, for the target region, is described by the NFW profile [34]:

$$\rho_{\mathrm{DM}}(r) = \frac{\rho_0}{\left(\frac{r}{r_s}\right)\left(1 + \frac{r}{r_s}\right)^2}\,, \tag{1}$$

---

[1]For instance, we recall that, as a rule of thumb, ICS processes upscatter the ambient photon energy from its initial low value $E_{\gamma}^{0}$ to a final value of up to $E_{\gamma} \approx 4\gamma^{2}E_{\gamma}^{0}$. Here $\gamma = E_{e}/m_{e}$ is the relativistic factor of the electrons and positrons. Hence a 10 GeV electron will produce a $\sim 0.15$ MeV hard X-ray when scattering off the CMB ($E_{\gamma}^{0} \approx 10^{-4}$ eV), or a $\sim 1.5$ GeV $\gamma$-ray when scattering off optical starlight ($E_{\gamma}^{0} \approx 1$ eV). For bremsstrahlung, the energy of the emitted photon peaks at a fraction of the initial energy of the $e^{\pm}$, typically between 1/10 and 1/2, depending on both the $e^{\pm}$ spectrum and on local conditions. See sections 2.1.2 and 2.1.3 for the full treatment.

where $r$ is radial distance from the GC. For the parameters $\rho_0$ and $r_s$ we use the values corresponding to the fitted NFW profile parameters (the central values) obtained in [35] for the baryonic model B2; they are similar to the ones used in [27, 33, 36].[2] In the following, we derive our main results using this NFW profile. However, for comparison we will also consider other choices, see sec. A.1.

As mentioned in the Introduction, for the considered weak-scale DM scenario, the main sources of photon signals from the GC region over the energy range of interest $(0.1\,\mathrm{MeV} \lesssim E_\gamma \lesssim 100\,\mathrm{GeV})$ are the prompt $\gamma$-ray emission and the emissions via Inverse Compton Scatterings and bremsstrahlung. As we will see, in some cases, the secondary components, especially the ICS, can become more important compared to the primary one over the considered photon energy range. Below we describe the production of these photon signals.[3]

### 2.1.1 Primary signal: Prompt $\gamma$-ray emission

DM annihilation into a given channel (e.g., $e^+e^-$, $\mu^+\mu^-$, $b\bar{b}$ or $W^+W^-$) gives rise to the prompt $\gamma$-ray flux which at the observer location can be computed as:

$$\frac{d\Phi_{\mathrm{prompt}}}{dE_\gamma d\Omega} = \frac{\langle \sigma v \rangle}{8\pi\, m_{\mathrm{DM}}^2} \frac{dN_\gamma}{dE_\gamma} \frac{J_{\Delta\Omega}}{\Delta\Omega}, \tag{2}$$

where $\langle \sigma v \rangle$ and $m_{\mathrm{DM}}$ are the velocity-averaged annihilation cross-section and the mass of the DM, respectively. The distribution $\frac{dN_\gamma}{dE_\gamma}(E_\gamma, m_{\mathrm{DM}})$ denotes the photon energy spectrum produced per annihilation in the considered annihilation channel. Such a spectrum is determined using the analytical expressions in [22] for $m_{\mathrm{DM}} < 5$ GeV and using the PPPC4DMID tools [37] for $m_{\mathrm{DM}} \geq 5$ GeV. Finally, the astrophysical $J$-factor for DM annihilation, $J_{\Delta\Omega}$, is defined for the observation region $\Delta\Omega$ as:

$$J_{\Delta\Omega} = \int_{\Delta\Omega} d\Omega \int_{l.o.s.} ds\, \rho_{\mathrm{DM}}^2(r(s,\theta)). \tag{3}$$

Here $s$ is the line-of-sight ($l.o.s.$) coordinate (with respect to the observer) which is related to the radial and angular distances from the GC, $r$ and $\theta$, as: $r = \sqrt{s^2 + r_\odot^2 - 2sr_\odot \cos\theta}$, with $r_\odot$ being the distance of the Earth from the GC [35].

In fig. 1, the blue curves show the prompt $\gamma$-ray fluxes discussed in eq. (2) (averaged over the observation region $\Delta\Omega$), for four benchmark values of the DM mass: $m_{\mathrm{DM}} = 1, 10, 10^2$ and $10^3$ GeV, considering the NFW DM profile (eq. 1) and the DM DM $\to \mu^+\mu^-$ annihilation channel with $\langle \sigma v \rangle$ set at the thermal value $\langle \sigma v \rangle = 3 \times 10^{-26}\,\mathrm{cm^3 s^{-1}}$.

### 2.1.2 Secondary signal: Inverse Compton ccatterings

We now move to consider the photon fluxes generated due to the ICS of DM-induced $e^\pm$ off the ambient photon bath, composed of mainly CMB, infrared (IR) and starlight (SL), collectively denoted as the Inter-Stellar Radiation Field (ISRF). The $e^-/e^+$ population produced by DM annihilations is written in terms of the source function as:

$$Q_e(E_e^S, r) = \frac{\langle \sigma v \rangle}{2\, m_{\mathrm{DM}}^2} \frac{dN_e}{dE_e^S} \rho_{\mathrm{DM}}^2(r), \tag{4}$$

---

[2]We checked that, using the NFW parameters tabulated in [1], our results (i.e., the DM annihilation signals and the corresponding constraints on DM annihilation) change at most by $\sim 20\%$.

[3]Apart from these, the production of another secondary photon signal from the In-flight annihilation (IfA) of DM induced positrons is also possible [21]. However, we checked that, in our considered photon energy range, and for our DM scenario, this flux is well suppressed compared to the other types of flux.

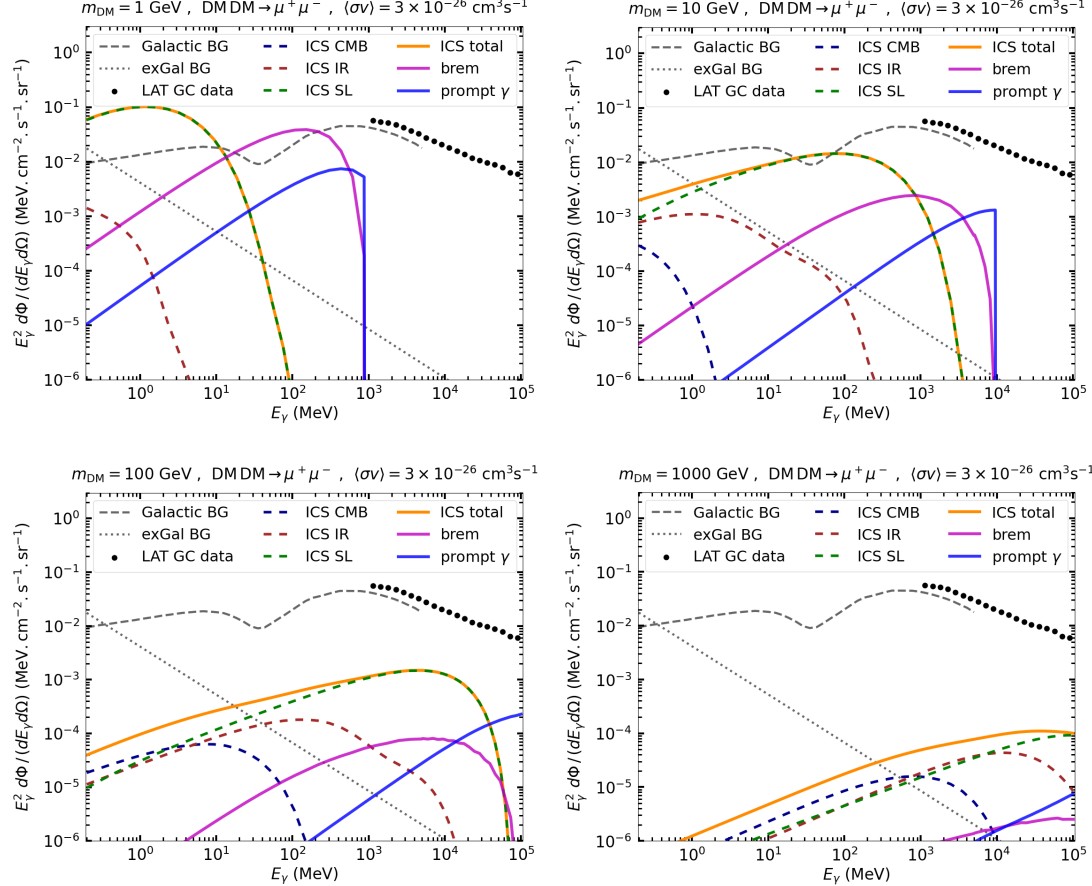

Figure 1: *Different types of **DM-induced photon fluxes** discussed in eqs. (2), (8) and (11), averaged over a disk of 10° around the GC. The four panels are for four different values of the DM mass: $m_{\rm DM} = 1, 10, 10^2$ and $10^3$ GeV. Here, for definiteness, we consider $\rm DM\, DM \to \mu^+\mu^-$ annihilations with $\langle\sigma v\rangle = 3 \times 10^{-26}$ cm³s⁻¹. In addition, different diffuse photon backgrounds and the FERMI-LAT GC γ-ray data are also shown (see the text for details). This figure shows that, in our energy range of interest, the secondary photon flux such as the ICS can be a very important component of the DM signal for different DM masses.*

where $\frac{dN_e}{dE_e^S}(E_e^S, m_{\rm DM})$ is the energy spectrum of $e^-/e^+$ sourced by DM annihilation into a given channel. Such spectra are estimated using the analytical expressions in [22] for $m_{\rm DM} < 5$ GeV and using the tools in [37] for $m_{\rm DM} \geq 5$ GeV.

The $e^\pm$, after being produced from DM annihilation, propagate through the galactic medium undergoing various effects, such as spatial diffusion, advection, convection, re-acceleration and radiative energy losses, and give rise to a steady state distribution [24,38,39]. Since we will be interested in a region of the galactic halo that does not cover the scale of the accretion region of the central black hole, we will assume that the effects of advection and convection can be neglected. On the other hand, near the region around the GC, for $e^\pm$ with energies above a GeV the effect of energy loss via various radiative processes becomes dominating over other processes, e.g. spatial diffusion and advection [21]. Also, many of the above-mentioned processes turn out to be more relevant for the propagation of the cosmic-ray nuclei rather than for the $e^\pm$ (see, e.g., [38,40] or [1] for a summary). We will therefore adopt here a simplified treatment by assuming that energy losses via various radiative processes dominate

the propagation of the DM-induced $e^{\pm}$. This assumption, referred to as the "on the spot" approximation, gives a semi-analytic way to solve the propagation of $e^{\pm}$ in the Galaxy. However, in Sec. 5.1 we will illustrate the effects caused by considering the full propagation of $e^{\pm}$, and see that the corresponding DM signals from our target region (and hence the limits on DM based on that) vary at most by a factor of few for the DM mass in the usual GeV–TeV range. A somewhat similar result was previously obtained in [41] and [7, 21, 42].

With the above-mentioned assumption, the spatial and energy distribution of the steady state $e^{\pm}$'s that give rise to the ICS flux is estimated as [7, 22, 23]:

$$\frac{dn_e}{dE_e}(E_e, \vec{x}) = \frac{1}{b_{\text{tot}}(E_e, \vec{x})} \int_{E_e}^{m_{\text{DM}}} dE_e^S \, Q_e(E_e^S, r),$$ (5)

where $b_{\text{tot}}(E_e, \vec{x})$ is the total energy loss rate of the $e^{\pm}$ (having an energy $E_e$) at a position $\vec{x}$ in the Galaxy. This $b_{\text{tot}}(E_e, \vec{x})$ includes energy losses of $e^{\pm}$ by various radiative processes, i.e. ICS off the ambient photon baths, synchrotron emission in the Galactic magnetic field, Coulomb interactions with the interstellar gases, ionization of the same gases and bremsstrahlung on the same gases. We refer the reader to [43] for their detailed expressions and just remind here that the energy losses through the Coulomb interaction, ionization and bremsstrahlung are more important when the $e^{\pm}$ energy $E_e$ is below a GeV, while for $E_e$ above a GeV, the losses due to ICS and synchrotron dominate.

Using the $e^+/e^-$ distribution obtained in eq. (5) one can estimate the total ICS emissivity as:

$$j_{\text{ICS}}(E_\gamma, \vec{x}(s, b, l)) = 2 \int_{m_e}^{m_{\text{DM}}} dE_e \sum_{i \in \text{ISRF}} \mathcal{P}_{\text{ICS}}^i(E_\gamma, E_e, \vec{x}) \frac{dn_e}{dE_e}(E_e, \vec{x}),$$ (6)

where the factor of 2 takes into account the contributions of both electrons and positrons. The coordinates $(s, b, l)$ correspond to respectively the $l.o.s.$ linear coordinate, latitude and longitude of a point with respect to the observer. The angular distance $\theta$ from the GC is given by $\cos\theta = \cos b \cos l$, and the vertical distance above the galactic plane $z = s \sin b$. The functions $\mathcal{P}_{\text{ICS}}^i$ denote the differential power emitted into photons with energy $E_\gamma$ due to the IC scatterings of an $e^+/e^-$ (having energy $E_e$) on each ambient photon bath $i$ in the ISRF (see [1, 3, 7] and Sec. A.2),

$$\mathcal{P}_{\text{ICS}}^i(E_\gamma, E_e, \vec{x}) = c \, E_\gamma \int d\epsilon \, n_i^{\text{ISRF}}(\epsilon, \vec{x}) \, \sigma_{\text{IC}}(\epsilon, E_\gamma, E_e),$$ (7)

where $i \to$ CMB, IR or SL. The differential photon number density $n_i^{\text{ISRF}}(\epsilon, \vec{x})$ corresponding to each ISRF component is the same one used in [22]. The quantity $\sigma_{\text{IC}}$ is the Klein-Nishina cross-section, obtained using the analytical expressions given in [1, 3, 7]. The limit of integration over $\epsilon$ in eq. (7) is determined by the kinematics of the IC scattering $1 \le q \le m_e^2/4E_e^2$, where $q \equiv \frac{E_\gamma m_e^2}{4\epsilon E_e(E_e - E_\gamma)}$. An illustration of $P_{\text{ICS}}^i$ (corresponding to the three ISRF components) as a function of $E_\gamma$ for different input $e^{\pm}$ energies can be found in figure 1 of [22].

Finally, the observable ICS photon flux (averaged over the observation region $\Delta\Omega$) is given by [22, 23]:

$$\frac{d\Phi_{\text{ICS}}}{dE_\gamma d\Omega} = \frac{1}{\Delta\Omega} \int_{\Delta\Omega} d\Omega \left[ \frac{1}{E_\gamma} \int_{l.o.s.} ds \, \frac{j_{\text{ICS}}(E_\gamma, \vec{x}(s, b, l))}{4\pi} \right].$$ (8)

For the spatial integral in the above equation, we cut the distribution of the $e^{\pm}$ density in the radial direction at $R = R_{\text{Gal}} = 20$ kpc, and in the vertical direction at $z = L_{\text{Gal}} = 4$ kpc, assuming this to be the size of the zone that keeps the $e^{\pm}$ confined.

In fig. 1 we plot the ICS photon fluxes (averaged over the observation region), for four values of the DM mass: $m_{\text{DM}} = 1, 10, 10^2$ and $10^3$ GeV, considering the DM DM $\to \mu^+ \mu^-$

annihilation channel with $\langle \sigma v \rangle = 3 \times 10^{-26}$ cm$^3$s$^{-1}$. We show contributions from different target photons (i.e., from CMB, IR, and SL) separately, together with the sum of all these contributions (the orange solid lines). Fig. 1 shows that, for different DM masses, the DM-induced ICS flux in principle can dominate over the other types of DM-induced signals in our energy range of interest, especially in the low energy regime.

### 2.1.3  Secondary signal: Bremsstrahlung

The same $e^\pm$ populations that produce the ICS flux can also give rise to the bremsstrahlung emission. In analogy with eq. (6), the bremsstrahlung emissivity can be expressed as:

$$j_{\text{brem}}(E_\gamma, \vec{x}(s, b, l)) = 2 \int_{m_e}^{m_{\text{DM}}} dE_e \, \mathcal{P}_{\text{brem}}(E_\gamma, E_e, \vec{x}) \frac{dn_e}{dE_e}(E_e, \vec{x}), \tag{9}$$

where the $e^\pm$ distribution $\frac{dn_e}{dE_e}(E_e, \vec{x})$ is given by eq. (5). The bremsstrahlung power emitted into photons with energy $E_\gamma$ due to the scattering of an $e^\pm$ of energy $E_e$ (with $E_e > E_\gamma$) is given by [43]:

$$\mathcal{P}_{\text{brem}}(E_\gamma, E_e, \vec{x}) = c \, E_\gamma \sum_i n_i(\vec{x}) \frac{d\sigma_i^{\text{brem}}}{dE_\gamma}(E_e, E_\gamma). \tag{10}$$

Here $n_i(\vec{x})$ describes the number density distribution of each of the gas species (ionic, atomic and molecular) and $\frac{d\sigma_i^{\text{brem}}}{dE_\gamma}(E_e, E_\gamma)$ is the corresponding differential scattering cross-section for bremsstrahlung. For all the details of these expressions, we refer again the reader to [43].

Finally, the observable photon flux due to bremsstrahlung (averaged over the observation region $\Delta\Omega$) is computed as:

$$\frac{d\Phi_{\text{brem}}}{dE_\gamma d\Omega} = \frac{1}{\Delta\Omega} \int_{\Delta\Omega} d\Omega \left[ \frac{1}{E_\gamma} \int_{l.o.s.} ds \, \frac{j_{\text{brem}}(E_\gamma, \vec{x}(s, b, l))}{4\pi} \right]. \tag{11}$$

The photon fluxes due to the bremsstrahlung emissions arising for different DM masses are shown in fig. 1 by the magenta curves. For our target region, bremsstrahlung emissions can dominate the DM signal in the intermediate part of our energy range of interest, especially when the DM mass is relatively smaller.

## 2.2  Background models and data

The total diffuse photon background towards our target region (i.e., the 10° cone around the GC) receives contributions from both galactic and extra-galactic origins.

The galactic photon backgrounds are obtained from [27] for the energy range $0.2 \, \text{MeV} \lesssim E_\gamma \lesssim 5 \, \text{GeV}$. This is the energy range that we use for the study of future MeV telescopes. The diffuse galactic background model consists of mainly four different astrophysical components. Three of them, namely: a bremsstrahlung component, a $\pi^0$ component and a high energy ICS component (ICS$_{\text{hi}}$), were computed with the GALPROP code [44] and fitted to the FERMI-LAT data at higher energies [45]. The fourth one, another ICS component (ICS$_{\text{lo}}$), was modeled as a power-law and fitted to COMPTEL and EGRET data at lower energies [21,46]. All these background models were presented in [21] for a region $|l| \leq 5°$, $|b| \leq 5°$ around the GC. Each of these models was then rescaled by [27] to the 10° cone region around the GC and presented in their figure 1. We adopt these models as our fiducial galactic photon backgrounds. On the other hand, for the extra-galactic component, we adopt the single power law model from [29], with their best-fit values of the normalization and the power law index as the fiducial parameters.

In each panel of fig. 1, the total fiducial galactic photon background (the sum of four components listed above) and the fiducial extra-galactic photon background are shown by the gray dashed and dotted lines, respectively. Along with these backgrounds, we also show in each plot with black points the FERMI-LAT $\gamma$-ray data towards the GC. These are taken from [47] for a $15° \times 15°$ region around the GC and presented in this figure by normalizing with respect to the corresponding solid angle.

## 3   Future MeV telescopes

We study the prospects of probing the DM signals in different near-future MeV space telescopes [14], such as AMEGO, E-ASTROGAM and MAST. Apart from these three, there are several other planned or proposed space-based MeV telescopes: for instance, COSI [26], GECCO [48], ADEPT [49], GRAMS [50] and PANGU [51]. These instruments have effective areas which are either smaller or of the same order of magnitude than the three above-mentioned instruments, and therefore are expected to provide comparatively smaller or similar sensitivities for the DM signal searches. We thus choose to focus on AMEGO, E-ASTROGAM and MAST, and give here a brief description of their main properties relevant for DM searches.

- ○ AMEGO: The All-sky Medium Energy Gamma-ray Observatory (AMEGO) is a proposed future space-based mission, poised to provide important contributions to multi-messenger astrophysics in the late 2020's and beyond. It can operate in two different modes, Compton scattering and pair-production modes, to achieve high sensitivity in a wide energy range 0.2 MeV to $\sim$ 5 GeV. The Compton mode is divided into two parts: untracked Compton and tracked Compton. By combining three different event classifications it can achieve an effective area of $\sim 500 - 1000 \, cm^2$ across the full energy range. A detailed overview of various instrumental details of AMEGO, e.g., effective area, energy resolution and angular resolution for different modes, can be found in Refs. [15, 16]. For our work we use the three effective areas discussed above and an energy resolution of 30% for the full range of energy over which it can operate. There is also a more recent proposal for an instrument, named AMEGO-X [17], with a similar concept.

- ○ E-ASTROGAM: E-ASTROGAM (or enhanced-ASTROGAM) is a satellite gamma-ray mission concept proposed by a wide international community for the late 2020's. Like AMEGO, it can operate in both Compton scattering and pair-production modes to achieve good sensitivity over a broad energy range 0.3 MeV to 3 GeV (the lower energy limit can be pushed to energies as low as 150 keV and 30 keV with improved tracker and calorimetric detections, respectively). Its effective area and energy resolution vary between $\sim 10^2 - 10^3 \, cm^2$ and $\mathcal{O}(1)\% - 30\%$, respectively, in the two modes over the energy range 0.3 MeV - 3 GeV. All the details related to its various instrumentation can be obtained from [18, 19]. In our case, we conservatively consider an energy resolution of 30% over its full energy range.

- ○ MAST: The Massive Argon Space Telescope (MAST) is another proposed future satellite-based telescope that plans to use a liquid Argon time projection chamber for $\gamma$-ray astronomy. Compared to the other future telescopes, it has a significantly larger effective area, which is estimated to be approximately $\sim 10^5 \, cm^2$ over the energy range $E_\gamma > 10$ MeV. For the instrumental details see [20]. We have considered an energy resolution of 30% for the full energy range of this telescope. The high sensitivity of MAST in searching MeV DM and primordial black hole (PBH) DM signals was discussed in [27, 32, 52] and [33, 53], respectively.

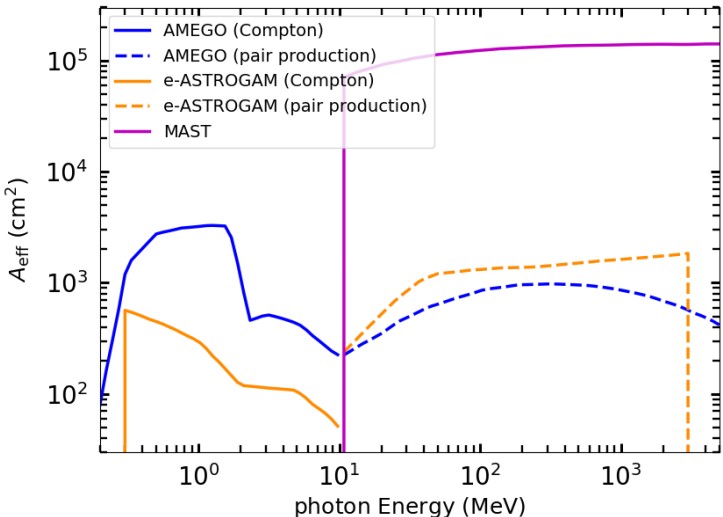

Figure 2: ***Effective areas*** *of the future telescopes* AMEGO, E-ASTROGAM *and* MAST, *for the range of the photon energy considered in this work.*

In fig. 2 the effective areas of the various considered future telescopes are shown for the range of the photon energy (0.2 MeV - 5 GeV) considered in this work. See also [27] for a more detailed summary of these telescopes.

# 4 Methodology

In this section we discuss the methodology adopted in this work to obtain the projections of the future MeV telescopes on the DM parameter space. To obtain these projections we consider the sum of ICS (8), bremsstrahlung (11) and prompt $\gamma$-rays (2) fluxes discussed in section 2 as the total DM-induced signal.

## 4.1 Conservative upper bounds

Before discussing the projections for the future MeV telescopes, we first derive some simple estimates of the possible upper bounds on the DM annihilation cross-section $\langle \sigma v \rangle$, based on existing $\gamma$-ray observations (in the range $0.2 \, \text{MeV} \lesssim E_\gamma \lesssim 100 \, \text{GeV}$).

For each of the considered annihilation channels, we obtain such estimates using the following approach. We require that, for a fixed $m_{\text{DM}}$, the total predicted DM photon signal, averaged over the $10°$ cone around the GC, does not exceed the total fiducial MeV photon background (the sum of the gray dashed and dotted lines from fig. 1, based on EGRET and COMPTEL data as discussed in sec. 2.2) at any energy bin in the range $0.2 \, \text{MeV} \lesssim E_\gamma \lesssim 5 \, \text{GeV}$, as well as the total signal, averaged over the $15° \times 15°$ region around the GC, does not exceed the FERMI-LAT data (the black points in fig. 1) at any energy bin in the range $5 \, \text{GeV} \lesssim E_\gamma \lesssim 100 \, \text{GeV}$. This leads to an upper bound on $\langle \sigma v \rangle$ for the given $m_{\text{DM}}$. The resulting constraints are drawn with a red solid line in figures 3 and 4 below.

## 4.2 Discovery reach of future MeV telescopes

In order to obtain the projected sensitivities or the discovery potentials of the MeV telescopes we take two approaches, discussed below.

### 4.2.1 Simple signal-to-noise ratio (SNR) method

In this case we take the simple approach presented in [32,33] and require the signal-to-noise ratio (SNR) over the observation time period to be larger than five, i.e.,

$$\frac{N_\gamma\big|_{\text{DM}}}{\sqrt{N_\gamma\big|_{\text{BG}}}} \geq 5 \,, \tag{12}$$

which leads to a $\sim 5\sigma$ projection on $\langle\sigma v\rangle$ for a given $m_{\text{DM}}$. Here $N_\gamma\big|_{\text{BG}}$ is the photon count of the total diffuse background (sum of the galactic and extra-galactic background components discussed in sec. 2), while $N_\gamma\big|_{\text{DM}}$ corresponds to the DM induced total photon signal for a given DM mass. The number of photon count over the energy range $[E_{\text{min}}, E_{\text{max}}]$ is defined as:

$$N_\gamma = t_{\text{obs}} \int_{E_{\text{min}}}^{E_{\text{max}}} dE_\gamma A_{\text{eff}}(E_\gamma) \int_{\Delta\Omega} d\Omega \frac{d\Phi}{dE_\gamma d\Omega} \,, \tag{13}$$

with

$$\frac{d\Phi}{dE_\gamma d\Omega} = \int dE'_\gamma R_\epsilon(E_\gamma, E'_\gamma) \frac{d\Phi}{dE'_\gamma d\Omega} \,, \tag{14}$$

where $\frac{d\Phi}{dE'_\gamma d\Omega}$ corresponds to either the DM signal or the diffuse background discussed in sec. 2. The function $R_\epsilon(E_\gamma, E'_\gamma)$ is a gaussian with mean $E'_\gamma$ and standard deviation $\epsilon(E'_\gamma)E'_\gamma$ that accounts for the finite energy resolution of the telescope [25,54]. The energy resolution $\epsilon$ and the effective area $A_{\text{eff}}$ of different MeV telescopes used here are discussed in sec. 3. The solid angle $\Delta\Omega$ corresponds to our ROI, a region of $10°$ radius from the GC. Note that the sensitivity of the projection scales with the observation time as $\sqrt{t_{\text{obs}}}$.

### 4.2.2 Fisher method

In this case we take an approach similar to the one presented in [27,36] involving the Fisher matrix method, employed previously in [55]. We define the vector $\vec{\theta}$ involving the signal and the background parameters:

$$\vec{\theta} = \left[ \Gamma^{\text{SIG}}, \theta_{\text{brem}}^{\text{BG}}, \theta_{\pi^0}^{\text{BG}}, \theta_{\text{ICS}_{\text{hi}}}^{\text{BG}}, \theta_{\text{ICS}_{\text{lo}}}^{\text{BG}}, \theta_{\text{e.g.}}^{\text{BG}} \right] \,, \tag{15}$$

where $\Gamma^{\text{SIG}}$ is the normalization of the DM signal (i.e., $\langle\sigma v\rangle$ for a fixed $m_{\text{DM}}$ and a fixed annihilation channel), while $\theta_i^{\text{BG}}$ denotes the rescaling of the normalization for each background component $i$ (representing the bremsstrahlung, $\pi^0$, $\text{ICS}_{\text{hi}}$, $\text{ICS}_{\text{lo}}$ and extra-galactic components) with respect to the fiducial one, discussed in sec. 2.2. The total differential photon flux is defined as:

$$\phi_{\text{tot}}(\vec{\theta}) = \frac{d\Phi^{\text{SIG}}}{dE_\gamma d\Omega}(\Gamma^{\text{SIG}}) + \sum_I \theta_I^{\text{BG}} \left\{ \frac{d\Phi_{\text{BG}}^I}{dE_\gamma d\Omega} \right\}_{\text{fiducial}} \,, \tag{16}$$

with $\frac{d\Phi}{dE_\gamma d\Omega}$ defined in eq. (14). Using these we define the Fisher matrix as:

$$\mathcal{F}_{ij} = t_{\text{obs}} \int_{E_{\text{min}}}^{E_{\text{max}}} dE_\gamma A_{\text{eff}}(E_\gamma) \int_{\Delta\Omega} d\Omega \left( \frac{1}{\phi_{\text{tot}}} \frac{\partial\phi_{\text{tot}}}{\partial\theta_i} \frac{\partial\phi_{\text{tot}}}{\partial\theta_j} \right)_{\vec{\theta}=\vec{\theta}_{\text{fiducial}}} \,, \tag{17}$$

where $\vec{\theta}_{\text{fiducial}}$ denotes the fiducial parameters with $(\Gamma^{\text{SIG}})_{\text{fiducial}}$ set to zero, i.e., considering the null hypothesis (no DM signal is present in the data) to be true. $\mathcal{F}$ here is a $6\times6$ symmetric matrix. The $2\sigma$ projected upper bound on the signal normalization $\Gamma^{\text{SIG}}$ is then defined as:

$$\Gamma^{\text{SIG}}_{\text{proj}} = 2 \sqrt{(\mathcal{F}^{-1})_{11}} \,. \tag{18}$$

Here, too, the sensitivity of the projection scales as $\sqrt{t_{\text{obs}}}$.

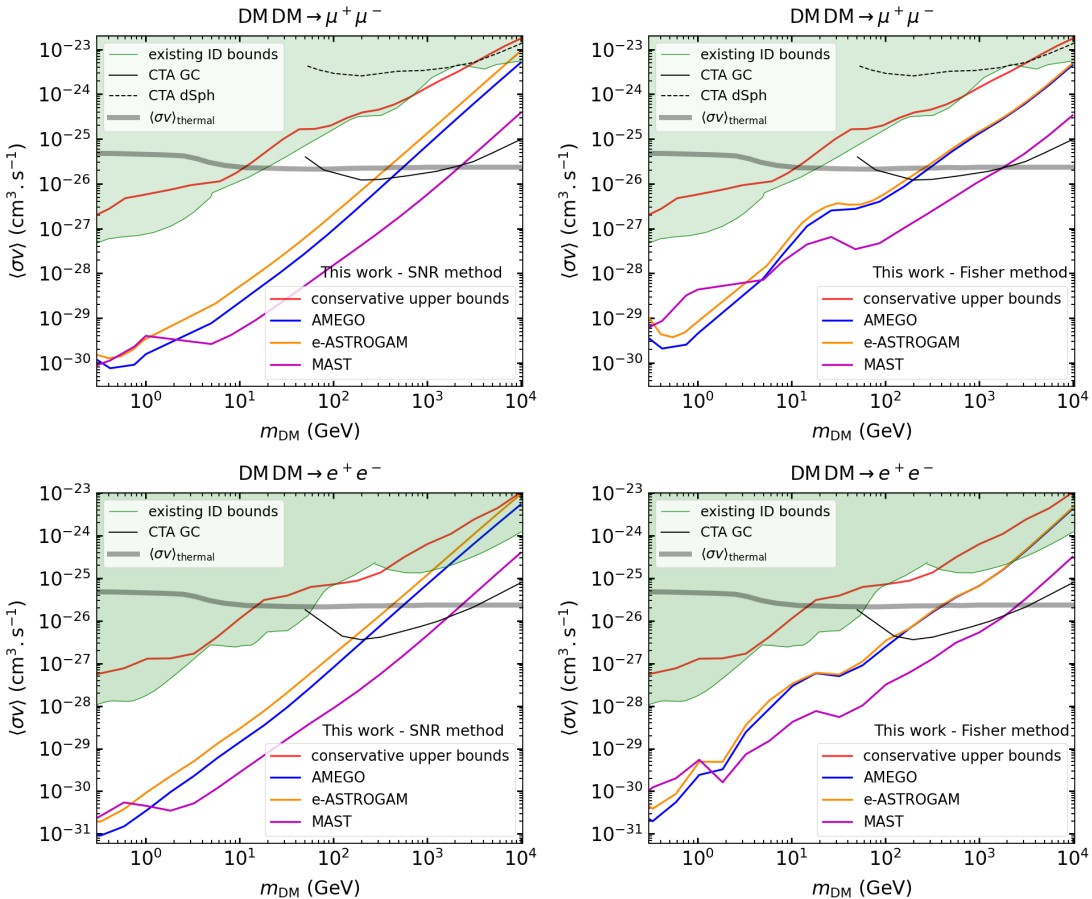

Figure 3: ***Upper bounds and projected sensitivities*** *on the annihilation cross-section* $\langle \sigma v \rangle$ *as a function of* $m_{\mathrm{DM}}$, *for* $\mu^+\mu^-$ *(upper panels) and* $e^+e^-$ *(lower panels) annihilation channels. Notice the different vertical ranges for different channels. For each channel, the projected sensitivities are shown considering the two statistical approaches discussed in sections* 4.2.1 *(left column) and* 4.2.2 *(right column). In each case, our conservative upper bounds are shown with a red solid line, and the existing bounds from the literature with green shaded areas (see the text for details). The projected sensitivities of the upcoming MeV telescopes* AMEGO, E-ASTROGAM *and* MAST *are shown by blue, orange and magenta curves, respectively, for an observation time of* $10^8$ *sec (*$\simeq 3$ *yrs). The solid and dashed black curves show the projections of* CTA *for the observations towards GC and dwarf galaxies (when available), respectively. Finally, the solid gray thin band in each plot indicates the value of* $\langle \sigma v \rangle$ *corresponding to the observed relic abundance for thermal DM, assuming s-wave annihilation* [56].

# 5 Results and discussion

In this section we present the bounds and the projected sensitivities of the future MeV telescopes on DM annihilation, derived in this work using the methodology outlined in sec. 4. The main results of this analysis are presented in fig. 3 (for $\mu^+\mu^-$ and $e^+e^-$ channels) and fig. 4 (for $b\bar{b}$ and $W^+W^-$ channels) for the usual weak-scale DM mass range spanning the GeV–TeV scale. We recall that our target of observation is the inner galactic region (see sec. 2) from where the secondary emission components such as the ICS flux are expected to be very high due to the high densities of the ISRF photons (especially the star light (SL)). This helps to increase the reach of the MeV telescopes for GeV–TeV DM. The high DM density that one ex-

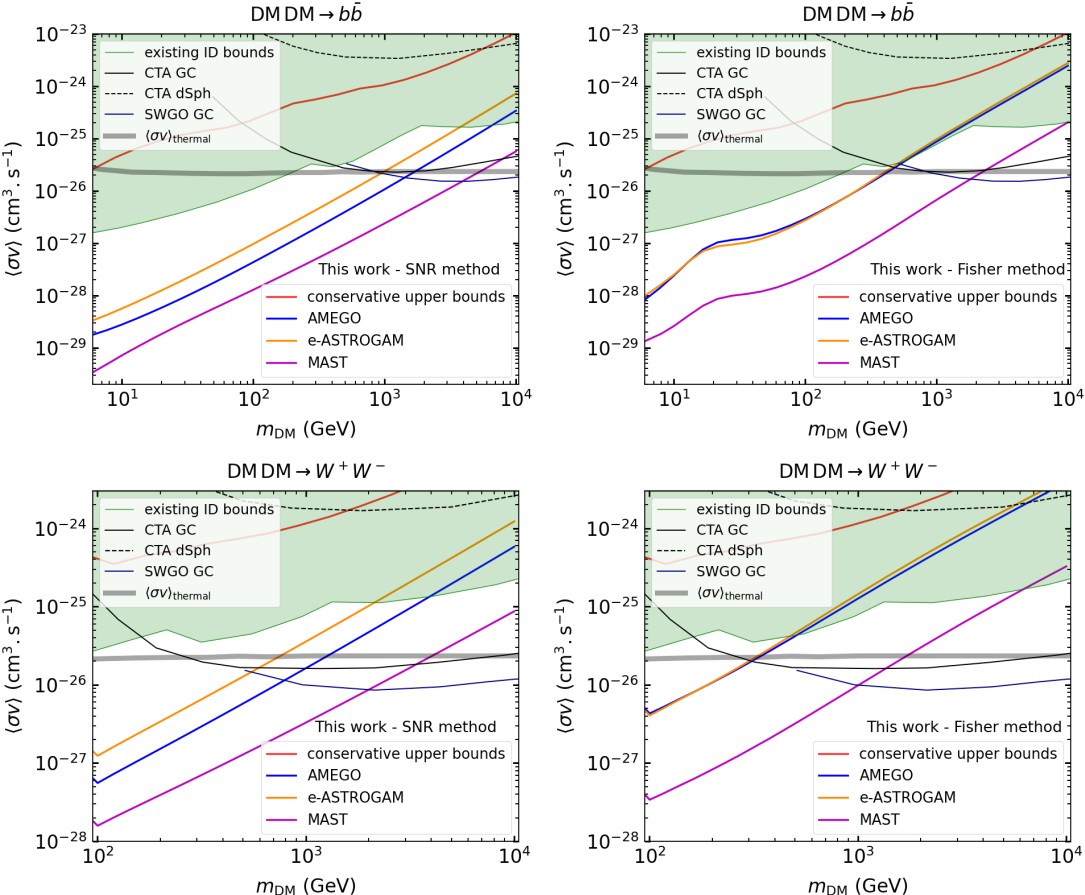

Figure 4: *Same as fig. 3, but considering $b\bar{b}$ (upper panels) and $W^+W^-$ (lower panels) annihilation channels. Notice the different horizontal and vertical ranges for different channels. The solid and dashed black curves show the projections of* CTA *for the observations towards GC and dwarf galaxies, respectively, and the dark blue curve shows the projection for* SWGO*, when available.*

pects in this region also helps. The different astrophysical ingredients used here are the ones discussed in sec. 2. In appendix A we discuss the variations of such astrophysical ingredients.

The solid red lines in figs. 3 and 4 show our estimated upper bounds on $\langle \sigma v \rangle$ as a function of $m_{\text{DM}}$, obtained in sec. 4.1 and corresponding to existing MeV-GeV $\gamma$-ray observations, essentially from COMPTEL, EGRET and FERMI-LAT.

Then, for each of the considered DM annihilation channels, we present the projected sensitivities of the upcoming MeV telescopes AMEGO, E-ASTROGAM and MAST with blue, orange and magenta curves, respectively. For each channel in figs. 3 and 4, the left and right panels correspond to the projections for different MeV telescopes estimated with the two statistical approaches discussed in sections 4.2.1 and 4.2.2, respectively. The observation time $t_{\text{obs}}$ is assumed to be $10^8$ sec ($\simeq$ 3 yrs) [32,33] which is a standard duration achievable by the future MeV telescopes considered here [15–20]. As pointed out earlier, the projected sensitivities scale with the observation time as $\sqrt{t_{\text{obs}}}$.

We compare our results with the existing bounds (shown by the green shaded areas in figs. 3 and 4) obtained from the literature. For a given annihilation channel, these bounds correspond to the most stringent upper limit on $\langle \sigma v \rangle$ (as a function of $m_{\text{DM}}$) that is compatible with all existing experimental observations. The existing bounds for the $\mu^+\mu^-$ (for $m_{\text{DM}} > 5$ GeV), $b\bar{b}$ and $W^+W^-$ channels are taken from [1]. For the $\mu^+\mu^-$ channel, this bound is

a combination of limits derived from CMB (assuming $s$-wave annihilation) [57, 58], dwarf galaxies $\gamma$-rays [59] and ANTARES neutrino [60] observations. On the other hand, for the $b\bar{b}$ and $W^+W^-$ channels these bounds are the convolution of limits obtained from dwarf gamma-rays [61], AMS anti-proton [62] and HESS GC [63,64] observations. The bounds for $\mu^+\mu^-$ and $e^+e^-$ for $m_{DM} < 5$ GeV are the combinations of limits from XMM-NEWTON $X$-ray [23] and CMB ($s$-wave) observations. For the $e^+e^-$ channel, the bound for $m_{DM} > 5$ GeV is taken from [65,66] and it is a convolution of the limits from AMS positron [67, 68], FERMI-LAT dwarfs [69] and HESS GC observations.

Our main result, conveyed by figs. 3 and 4, is that, while the upper bounds (the red curves) on $\langle \sigma v \rangle$ obtained based on the existing $\gamma$-ray observations in MeV–GeV range remain mostly within the combined Indirect Detection exclusions (the green shaded regions) discussed above, the future MeV telescopes have the potential to probe large regions of the DM parameter space that are yet unexplored.

Considering the projected sensitivities estimated with the Fisher matrix method, such regions for the $b\bar{b}$ and $W^+W^-$ annihilation channels extend up to a few hundreds of GeV (for AMEGO and E-ASTROGAM) or up to a few TeV (for MAST), for thermal DM. For the leptonic channels the forecast sensitivities turn out to be even better. From fig. 3 we see that, for the $e^+e^-$ and $\mu^+\mu^-$ channels, the DM parameter space that remains unconstrained but lies within the reach of the future MeV telescopes can be extended up to the TeV or even the multi-TeV mass scale, for thermal DM. Such a statement is in general true for all the MeV telescopes considered here, although MAST is expected to provide a comparatively better sensitivity. Considering a 100 GeV DM annihilating to $\mu^+\mu^-$, the Fisher forecasts for AMEGO and E-ASTROGAM reach a value of $\langle \sigma v \rangle$ which is almost a factor of 30 below the present experimental bound, while for MAST this value can go down by an order of magnitude more.

Taking the rather simplified approach of the signal-to-noise ratio (SNR) method, the estimated sensitivities of the future telescopes extend even further. In figs. 3 and 4 this can be seen more prominently for $m_{DM}$ up to the TeV scale for the $\mu^+\mu^-$ and $e^+e^-$ channels, and over almost all the DM mass range for the $b\bar{b}$ and $W^+W^-$ channels.

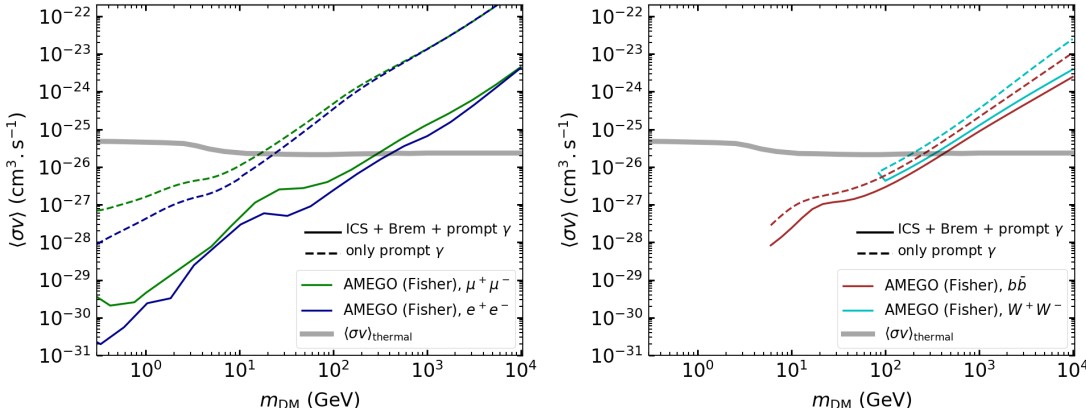

Figure 5: *Illustration of the **importance of considering the secondary signals** of DM in the case of the leptonic annihilation channels. The comparison between the Fisher projections (for AMEGO with $t_{obs} = 10^8$ sec) obtained considering and not considering the secondary signals are shown for the $\mu^+\mu^-$, $e^+e^-$ (left panel) and the $b\bar{b}$, $W^+W^-$ (right panel) channels. The colored solid curves are the projections obtained considering the secondary signals and are the same as the ones presented in figs. 3 and 4, while the dashed curves show the projections obtained not considering the secondary signals.*

Note that, compared to the simplified SNR approach, the Fisher method in general provides a more realistic estimate for signal normalization by accounting for statistical uncertainties in the backgrounds as well as the differences in the spectral shapes. In addition, it also takes into account correlations between background and signal. On the other hand, the SNR approach relies mainly on the comparison between the count of the number of photons from the signal to that from the total background (see Eq. 12). Hence, for a fiducial background model, the SNR is proportional to the signal number count which is the integration of the signal spectrum over the entire energy range of interest (e.g., ∼ 0.2 MeV – 5 GeV for the MeV telescopes like AMEGO). Now, in the cases where the total photon signal fills the entire photon energy window of interest (from low to high energies), such an integration of the signal spectrum over the energy leads to a large signal count and thus a stronger projection on the signal normalization compared to that obtained with the Fisher approach. For example, note from fig. 1 that such a situation for the leptonic annihilation channels like $\mu^+\mu^-$ is more prominent for $m_{\mathrm{DM}}$ in the range 1 – 100 GeV, thanks to the inclusion of low energy secondary photon spectra (mainly the ICS signals). As a result, for these mass scales, the SNR method yields better projections compared to the Fisher method, as can be seen by comparing the left and right panels of fig. 3. As one moves towards the TeV mass scale, the corresponding contribution of the total signal becomes less efficient in filling the full energy window due to its fall at lower energies (even after including the secondaries) (see for the case $m_{\mathrm{DM}} = 1$ TeV in fig. 1). Hence, the total photon count also gets smaller, leading to a weaker SNR projection (which can be even weaker than that obtained with the Fisher method), as can be seen again by comparing the left and right panels of fig. 3. On the other hand, for the hadronic channels, since the total signal (primary + secondaries) is softer and hence covers the whole energy range interest in a somewhat uniform way for almost all DM masses (see, for example, the two plots for the $W^+W^-$ channel in fig. 10), the SNR method provides a better projection for all DM masses (as can be seen by comparing the left and right panels of fig. 4).

Among the different MeV telescopes considered here, the forecast sensitivity of MAST turns out to be comparatively better, mainly because of the large effective area estimated for this telescope (see fig. 2). As can be seen from figs. 3 and 4, using such an instrument it could be possible to probe in the future a yet unconstrained thermally-annihilating DM model even at the scale of $\mathcal{O}(10)$ TeV.

MeV telescopes provide a good sensitivity for *leptonic* annihilation channels (like $\mu^+\mu^-$ and $e^+e^-$) mainly because these channels give rise, through cascades, to copious pairs of $e^{\pm}$, which in the case of usual GeV–TeV scale DM produce strong secondary signals as MeV $\gamma$-rays. This is illustrated in fig. 5 (left panel), where the Fisher forecast projections for AMEGO obtained considering the total DM signal are compared to those obtained considering only the prompt $\gamma$-ray signals. The prospects for the leptonic channels improve by up to two orders of magnitude when the secondary emission is considered, which shows clearly the importance of including such signals for weak-scale DM. For *hadronic* channels (right panel of fig. 5) the improvement is present but much reduced.

Apart from the MeV $\gamma$-ray telescopes, there are various planned and proposed ground-based high-energy $\gamma$-ray telescopes such as CTA [70] and SWGO [71], which are also expected to start operating in the forthcoming years. These telescopes are sensitive to $\gamma$-rays above ∼ 50 GeV. For example, the operating energy range for CTA is 50 GeV – 50 TeV, while for SWGO it is 100 GeV – 1 PeV. As a result, they can also play an important role in probing the annihilations of relatively heavy DM. In figs. 3 and 4 we show the projected sensitivities of CTA and SWGO and compare them with those obtained for the MeV telescopes in this work. The solid and dashed black curves show the projections of CTA for the observations towards the GC [72] and the dwarf spheroidal (dSph) galaxy Ursa Major II [73], respectively, for 500 hrs of observation time (that corresponds to ∼ 5 yrs of runtime [74]). The dark blue solid curves show the

projections of SWGO for the observation towards the GC for 10 yrs [75]. The projections are shown here only for those channels that are available in the corresponding literature. As expected, among different targets, the highest sensitivities of these telescopes are achieved for the observation towards the GC. The high sensitivity of these telescopes for the energetic $\gamma$-rays or correspondingly for heavy DM is associated to their large effective areas based on the ground. As can be seen from figs. 3 and 4, while these high energy $\gamma$-ray telescopes may have a comparatively better sensitivity for a DM mass above a few 100 GeV (when compared to AMEGO or E-ASTROGAM) or a few TeV (when compared to MAST), below such mass ranges the latter class of instruments provide the strongest sensitivities so far in probing DM signals. Therefore, one can say that the future space-based MeV $\gamma$-ray telescopes will very efficiently complement the ground-based high energy $\gamma$-ray instruments in the indirect searches for weak-scale DM.

Note that in our work the projected sensitivities to DM annihilation from different future MeV telescopes are obtained using the two methods (SNR and Fisher) described in sections 4.2.1 and 4.2.2. These methods are efficient in obtaining the projections for DM in the sense that they focus on comparing the putative DM signal to the astrophysical backgrounds by taking into account the statistical fluctuations in the background model as well as the correlations between backgrounds and signal (applicable for the Fisher analysis). However, they do not account directly for the systematic uncertainties in the background model itself, which can be dominant over the statistical fluctuations and are expected to be at the level of 15% for the considered backgrounds [25, 76]. Since our analyses (based on SNR and Fisher forecasts) use the square-root of the background photon flux, the addition of a systematic uncertainty to our fiducial background at a level of 15% is expected to weaken our projections by $\sim$ 7% only (see also [27] for a similar discussion). In order to verify this for both the methods, we re-estimate some of the projections shown in figs. 3 and 4 by conservatively enhancing each of the fiducial background components (discussed in section 2.2) by a factor of 2 (at all photon energies), and verified that indeed the projections are weakened *maximally* by $\sim$ 40% compared to the fiducial ones. For a somewhat similar quantitative discussion about this, see for example [77].

## 5.1 Discussion on the effects of propagation of $e^{\pm}$ in the Galaxy

In this sub-section we discuss the effects of considering the full propagation of DM-induced $e^{\pm}$ in the Galaxy. In fig. 6 we show these effects on the DM induced photon signal for two values of the DM mass, $m_{\text{DM}} = 1$ GeV and 1 TeV, considering the $\mu^+\mu^-$ annihilation channel. The red solid curves correspond to the sum of DM-induced ICS, bremsstrahlung and prompt $\gamma$-ray fluxes shown in fig. 1, where the $e^{\pm}$ distribution required to estimate the secondary signals in the Galaxy were obtained using eq. (5). The red broken curves, on the other hand, correspond to the sum of similar quantities, but the $e^{\pm}$ distribution here are obtained by solving the full Galactic propagation that includes different processes, such as spatial diffusion, advection, convection, re-acceleration, radiative energy losses and various nuclear processes; see for example eq. 2.1 of [38] or [39]. In order to solve this we use the package DRAGON2 [38, 39].[4] The magnetic field is assumed to be the default model implemented in DRAGON2. The DM profile is considered to be the same one used in fig. 1.

---

[4] DRAGON2 (used here in the context of estimating the galactic DM signals) is similar in scope to GALPROP (used in the literature to estimate different Galactic backgrounds mentioned above). Both are designed to cover most of the relevant processes involving galactic cosmic-rays and their secondary products, over a wide energy range. DRAGON2 and GALPROP implement similar algorithms to solve the galactic propagation equation of charged particles such as $e^{\pm}$'s, see [38] and [44, 78]. Moreover, the energy-loss processes, ISRF models, gas maps and various nuclear processes implemented in DRAGON2 are quite similar to the ones used in GALPROP. See [38] for a detailed discussion about these.

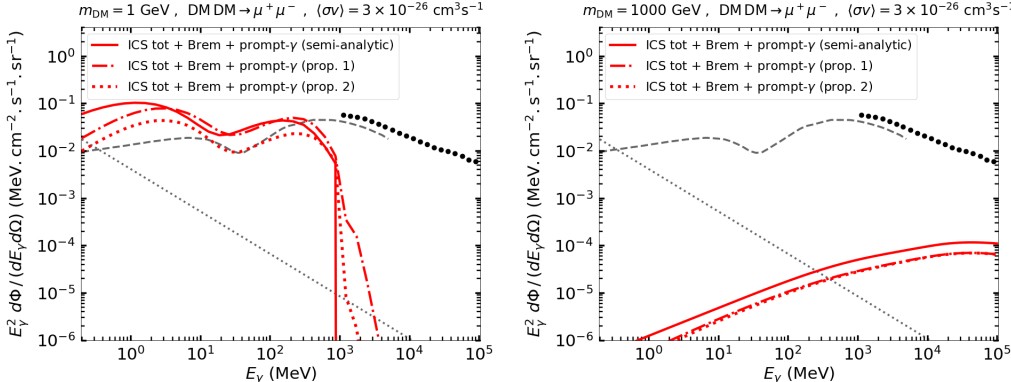

Figure 6: **Effects of the propagation** of $e^{\pm}$. The red solid curves correspond to the total photon flux (ICS + brem + prompt-$\gamma$) shown in fig. 1, where the secondaries are obtained with the $e^{\pm}$ distribution resulting from the semi-analytic approach (Eq. 5). The red broken curves show the same total photon flux, now computed with the $e^{\pm}$ distribution in the Galaxy resulting from solving the full propagation equation with DRAGON2 [38, 39]. In particular, the red dashed-dotted and dotted curves correspond to the two different propagation models discussed in the text. The backgrounds (gray lines) and the FERMI data (black points) are shown for illustration and are the same as the ones presented in fig. 1.

The red dashed-dotted and dotted curves in fig. 6 correspond to two Galactic propagation models, named here as "prop. 1" and "prop. 2", respectively. These models have been used in the past for the study related to the secondary photon emissions towards the GC.

**Model "prop. 1":** it refers to the "model z04LMS" from [79] (see their Table 1) which was adopted in [45] to estimate the different Galactic photon backgrounds (i.e., bremsstrahlung, $\pi^0$ and ICS components) mentioned in section 2.2 and considered in the present work. In such model the spatial diffusion is parameterized as a function of the propagating particle rigidity ($\rho$) as $D(\rho) = D_0 \beta^{\eta} (\rho/4 \, \mathrm{GV})^{\delta}$, with $\beta$ as the dimensionless particle velocity and $\eta = 1$. The values of $D_0$ and $\delta$ are $5.8 \times 10^{28} \, \mathrm{cm}^2 \mathrm{s}^{-1}$ and 0.33, respectively. The Alfvén velocity (related to the re-acceleration of the particles) is $v_A = 30 \, \mathrm{km/s}$. The size of the diffusion zone (beyond which the $e^{\pm}$ escape freely) is the same one mentioned in section 2.1 and used in our other results (i.e., $R_{\mathrm{Gal}} = 20 \, \mathrm{kpc}$ and $L_{\mathrm{Gal}} = 4 \, \mathrm{kpc}$).

**Model "prop. 2":** it refers to the "model A" adopted in [80] (see their Table 2) to estimate the flux for the above-mentioned Galactic backgrounds. This model was used in [27] to rescale the Galactic backgrounds to the region $10°$ cone around the GC which is our target ROI. For this model, $R_{\mathrm{Gal}} = 20 \, \mathrm{kpc}$, $L_{\mathrm{Gal}} = 4 \, \mathrm{kpc}$, $D_0 = 5.0 \times 10^{28} \, \mathrm{cm}^2 \mathrm{s}^{-1}$, $\eta = 0$, $\delta = 0.33$, $v_A = 32.7 \, \mathrm{km/s}$. In addition, this model includes a somewhat realistic convective wind velocity in terms of its gradient perpendicular to the Galactic plane, $dv/dz = 50 \, \mathrm{km/s/kpc}$. Notice that Model "prop. 2" is actually very similar to "prop. 1" for relativistic $e^{\pm}$; the only difference is that "prop. 2" includes a convective wind which the "prop. 1" does not.

Fig. 6 shows that, considering the full effects of propagation of $e^{\pm}$ in the Galaxy, the total DM signal across the GeV–TeV DM mass range can be suppressed *at most* by a factor of few (depending on the propagation models) in the entire photon energy range of interest.

To show this more prominently for the full DM mass range considered, we show in Fig. 7 the variations in the upper-bounds and the projections of the MeV telescopes in the $\langle \sigma v \rangle - m_{\mathrm{DM}}$ plane due to the effects of the Galactic propagation of $e^{\pm}$. For definiteness, this figure is shown only for the $\mu^+ \mu^-$ annihilation channel and for the MeV telescope AMEGO. The left and right panels correspond to the two statistical approaches discussed in sections 4.2.1 and

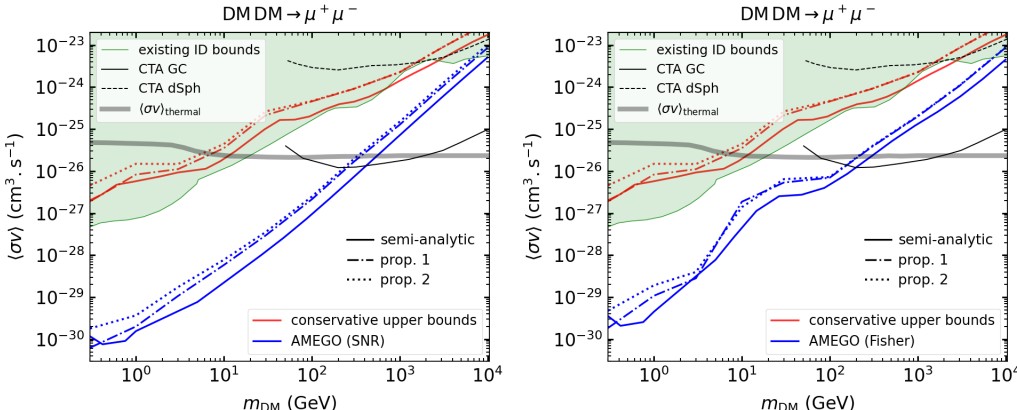

Figure 7: *Effects of the propagation of $e^{\pm}$ on the estimated bounds and projections are shown for the $\mu^+\mu^-$ annihilation channel. The projections are shown for* AMEGO *and are obtained using the two statistical approaches discussed in sections 4.2.1 (left panel) and 4.2.2 (right panel). The red and blue solid curves are respectively the same bounds and projections shown in fig. 3 and correspond to the case where the $e^{\pm}$ distribution in the Galaxy is obtained using the semi-analytic method described by Eq. (5). The red and blue broken curves, on the other hand, show the results obtained with the full propagation of $e^{\pm}$ in the Galaxy, for the two models discussed in the text.*

4.2.2, respectively. The solid red and blue curves refer to respectively the same bounds and projections shown in fig. 3 and correspond to the case where the $e^{\pm}$ distribution in the Galaxy is obtained using the semi-analytic method described by Eq. 5. On the other hand, the dashed-dotted (dotted) red and blue curves show the results estimated involving the full propagation of $e^{\pm}$ in the Galaxy with the propagation model "prop. 1" ("prop. 2") discussed above.

From Fig. 7 one can see that including a detailed and involved numerical simulation of the propagation of $e^{\pm}$ in the Galaxy under some commonly used propagation models (that are used to estimate the Galactic photon backgrounds towards the GC) the bounds and the projections of MeV telescopes on the weak-scale DM parameter space can be weakened *maximally* by a factor $\sim 2-3$ depending on the DM mass. Hence, the discussion of this section indeed shows that the conclusions based on the results obtained adopting the semi-analytic approach to estimate the Galactic distribution of DM induced $e^{\pm}$ remain the same.

Using propagation models different than the "prop. 1" or "prop. 2" used here is expected to change not only the DM induced signals but also to modify the Galactic backgrounds (composed of secondary radiations) in a similar fashion, leaving the constraints on DM annihilation in the same ballpark. For example, assuming a propagation scenario that enhances the signal flux, say, by a factor of a few (like in the case of the semi-analytic approach), one may expect the background flux to be also modified in a similar way over the energy range of interest. Since the analysis for obtaining the projections on DM (e.g., the signal-to-noise ratio in Eq. (12)) relies on the comparison of the signal to the fluctuation in the background flux (appearing as the square-root), the effects of both enhancements are expected to mitigate each other (more precisely, actually, the signal-to-noise ratio should be enhanced slightly, making the projections on the DM annihilation slightly stronger). There is also a possibility of translating such variations into the systematics. Hence, in order to be less dependent on different propagation parameters and to keep full control of our computations, we adopted the semi-analytic approach described in sec. 2.1 to obtain our main results (presented in figs. 3 and 4), and leave a more detailed numerical study involving different propagation models self-consistently applied to signals as well as to backgrounds for future work.

# 6 Conclusions

In this work, we have explored the sensitivity of future MeV telescopes to the photon signals produced by the annihilation of weak-scale DM particles, focusing in particular on the galactic center region. The central idea of our study is that these telescopes will be sensitive to the low-energy secondary emissions (essentially ICS and bremsstrahlung $\gamma$-rays produced by the DM-induced electrons and positrons) and will therefore be able to complement the high-energy $\gamma$-ray experiments, which are instead insensitive to the DM prompt emission. We focused on three representative planned experiments (AMEGO, e-ASTROGAM and MAST) and on a few representative DM annihilation channels. We adopted for definiteness a NFW DM galactic distribution, a streamlined treatment of the galactic propagation of $e^{\pm}$ and standard assumptions for the astrophysical environment (but we investigate in detail the impact of varying all these ingredients in section 5.1 and in Appendix A). We used two forecast approaches: a simple signal-over-noise ratio criterion and the more refined Fisher matrix method.

Our results are very promising. We find that, thanks to the fact that the secondary emissions significantly enhance the MeV signals of weak-scale DM annihilations, the MeV telescopes will explore a wide area of the $m_{DM} - \langle \sigma v \rangle$ parameter space. Figs. 3 and 4 summarize our main results. We find that an experiment like MAST, for an observation time of about 3 years, will be able to probe thermally-annihilating DM up to a few TeV's of mass, the details depending on the annihilation channel and the analysis method. This is comparable to the reach of future high-energy telescopes such as CTA and SWGO. For lower masses, the future MeV telescopes could be sensitive to DM annihilation cross sections that are 2 to 3 orders of magnitude smaller than the current bounds, again with the details depending on the channel and the analysis.

Summarizing, it is remarkable that MeV telescopes will be able to probe TeV DM, possibly competing with TeV telescopes, because of the significant leverage of secondary emissions. This adds an important tool in the continuing quest for the indirect detection of weak-scale DM.

# Acknowledgments

The authors acknowledge useful discussions with HaoYu Xie. They acknowledge the hospitality of the Institut d'Astrophysique de Paris (IAP) where part of this work was done. M.C. also acknowledges the hospitality of the Flatiron Institute of the Simons Foundation and New York University.

**Funding information**   CNRS grant *DaCo: Dark Connections*; research grant *DaCoSMiG* from the 4EU+ Alliance (including Sorbonne Université); Institut Henri Poincaré (UAR 839 CNRS-Sorbonne Université) and LabEx CARMIN (ANR-10-LABX-59-01).

# A   Impact of astrophysical uncertainties

## A.1   DM halo profile

Fig. 8 shows how our results, i.e. the bounds (top panel) and projections (bottom panels), vary with different choices for the galactic halo DM profile. The colored solid curves correspond to the NFW profile (eq. (1)) with the parameters $\rho_0$ and $r_s$ corresponding to the fitted parameters (the central values) obtained in [35] for the baryonic model B2, as used in all other results of this work. The dashed and dashed-dotted curves show, respectively, the results obtained using

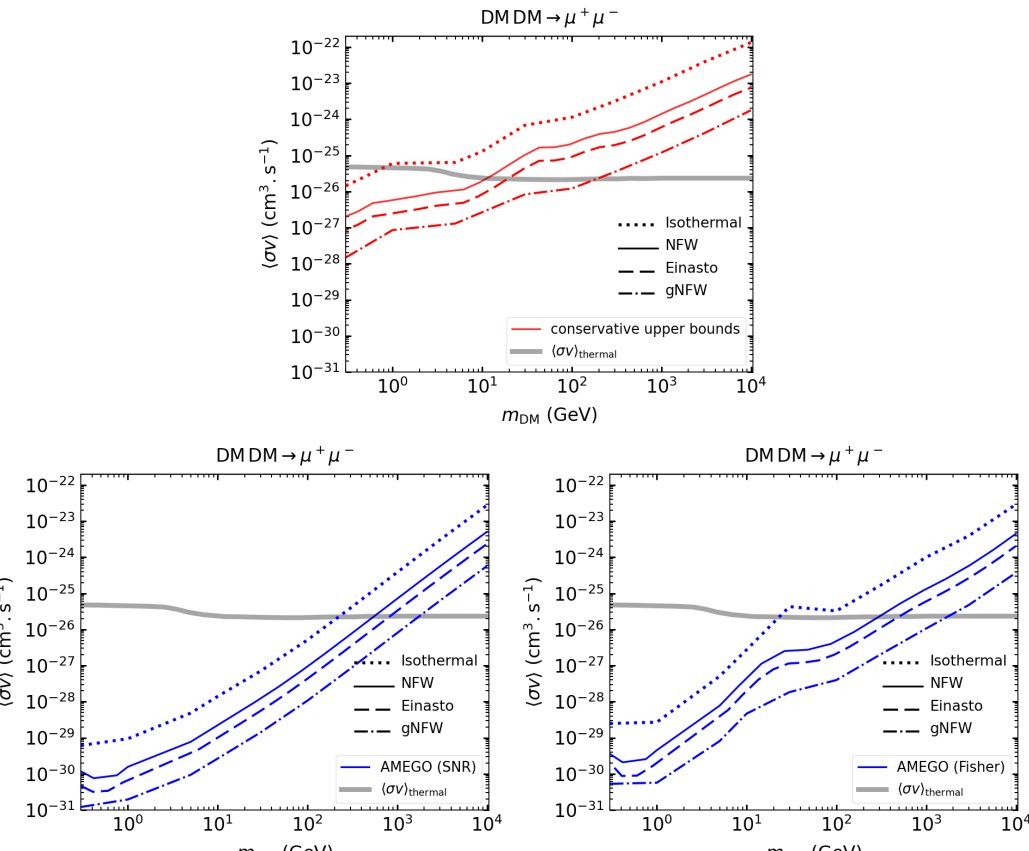

Figure 8: *Comparison of the bounds (upper panel) and projections for* AMEGO *(bottom panels) obtained with* **different galactic DM halo profiles**. *The projections are shown for the two statistical approaches discussed in sections 4.2.1 (bottom left) and 4.2.2 (bottom right). The colored solid curves correspond to the NFW profile and are the same as the ones shown in fig. 3. The dashed, dashed-dotted and dotted curves correspond to Einasto, generalized NFW (gNFW) and Isothermal profiles, respectively. See the text for details.*

the Einasto profile [81]

$$\rho_{\text{DM}}^{\text{Ein}}(r) = \rho_0 \exp\left\{-\frac{2}{\alpha}\left(\left(\frac{r}{r_s}\right)^\alpha - 1\right)\right\}, \tag{A.1}$$

and a generalized NFW (gNFW) profile

$$\rho_{\text{DM}}^{\text{gNFW}}(r) = \frac{\rho_0}{\left(\frac{r}{r_s}\right)^\gamma \left(1 + \frac{r}{r_s}\right)^{3-\gamma}}. \tag{A.2}$$

The corresponding profile parameters are set to their central values obtained in the fit in [35] using the same baryonic model B2 used for the NFW profile. The values of $\alpha$ and $\gamma$ corresponding to the Einasto and gNFW profiles are $\alpha = 0.18$ and $\gamma = 1.3$, respectively. In fig. 8 we also show, with dotted curves, the results obtained using a truncated isothermal (cored) profile

$$\rho_{\text{DM}}^{\text{Iso}}(r) = \frac{\rho_0}{1 + \left(\frac{r}{r_s}\right)^2}, \tag{A.3}$$

with the corresponding parameters tabulated in [1].

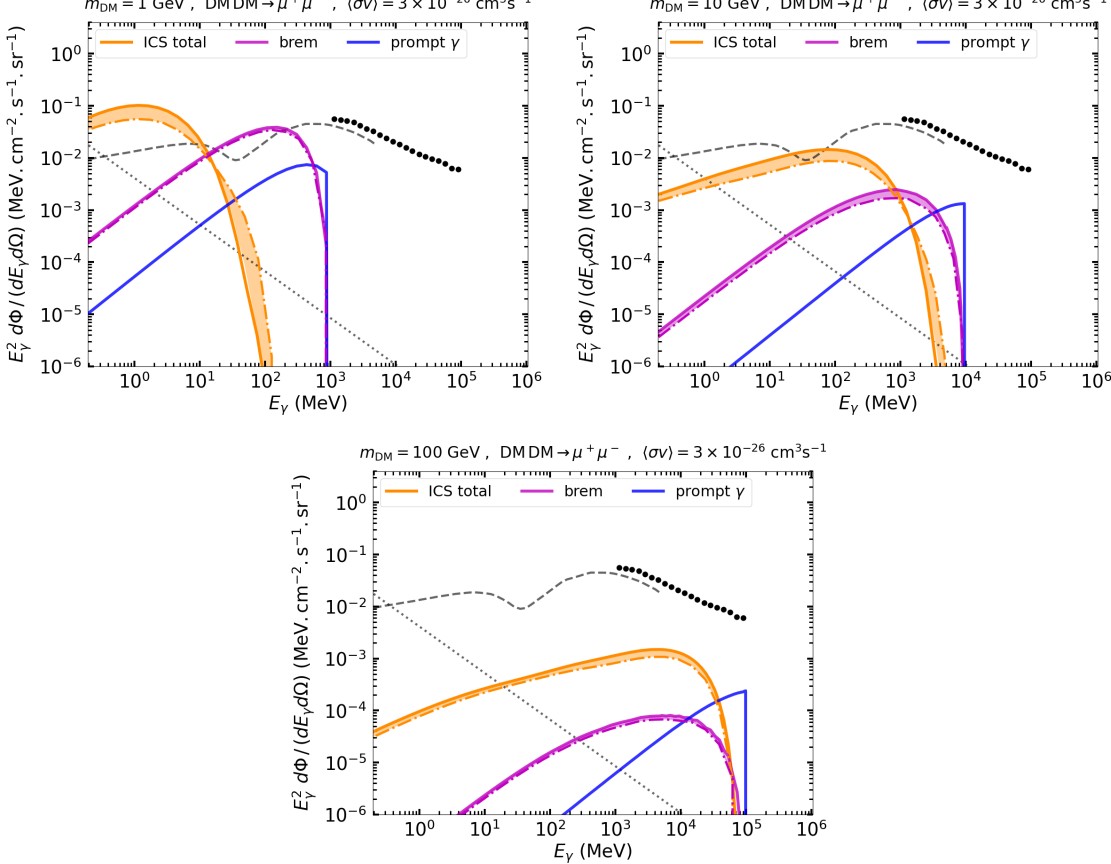

Figure 9: *DM-induced photon signals considering **different ISRF models**. The results are shown for $m_{\mathrm{DM}}$ = 1, 10 and 100 GeV considering the $\mu^+\mu^-$ annihilation channel. The dashed-dotted curves (for the ICS and bremsstrahlung signals) are obtained considering the ISRF model "Porter2006" (see [38] for a detailed discussion). The solid curves on the other hand correspond to the ISRF model discussed in sec. 2.1.2 and are the same as the ones shown in fig. 1.*

Form fig. 8 we see that, while considering Einasto and gNFW profiles the bounds and projections presented in fig. 3 can strengthen by a factor of 2 and by an order of magnitude, respectively, considering the Isothermal profile they loose by a factor of 10.

## A.2 ISRF model

In figs. 9 and 10 we show the level of variation in the DM induced secondary fluxes due to two different choices for the ISRF model. We show such variations for different values of the DM mass and for different annihilation channels. Fig. 9 shows these variations for the $\mu^+\mu^-$ channel considering three values of the DM mass, $m_{\mathrm{DM}}$ =1, 10 and 100 GeV. Fig. 10 shows the results for other channels and other $m_{\mathrm{DM}}$ values: $DM\,DM \rightarrow e^+e^-$, $m_{\mathrm{DM}} = 1$ GeV (*top left*); $DM\,DM \rightarrow e^+e^-$, $m_{\mathrm{DM}} = 100$ GeV (*top right*); $DM\,DM \rightarrow b\,\bar{b}$, $m_{\mathrm{DM}} = 10$ GeV (*middle left*); $DM\,DM \rightarrow b\,\bar{b}$, $m_{\mathrm{DM}} = 100$ GeV (*middle right*); $DM\,DM \rightarrow W^+W^-$, $m_{\mathrm{DM}} = 200$ GeV (*bottom left*); $DM\,DM \rightarrow W^+W^-$, $m_{\mathrm{DM}} = 1$ TeV (*bottom right*). These two figures serve as the representative of all the different DM annihilation scenarios considered in the present work. In each of these plots, the solid curves correspond to the ISRF model discussed in sec. 2.1.2 (which is used in the main results) and are the same as the ones shown in fig. 1 (for the $\mu^+\mu^-$ channel). On the other hand, the dashed-dotted curves correspond to the ISRF model

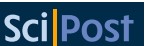

**Figure 10:** *Same as Fig. 9, but considering other annihilation channels and other DM masses: DM DM → e⁺e⁻, $m_{\mathrm{DM}} = 1$ GeV (top left); DM DM → e⁺e⁻, $m_{\mathrm{DM}} = 100$ GeV (top right); DM DM → b b̄, $m_{\mathrm{DM}} = 10$ GeV (middle left); DM DM → b b̄, $m_{\mathrm{DM}} = 100$ GeV (middle right); DM DM → W⁺W⁻, $m_{\mathrm{DM}} = 200$ GeV (bottom left); DM DM → W⁺W⁻, $m_{\mathrm{DM}} = 1$ TeV (bottom right).*

"Porter2006" (see [38] for a detailed discussion) that uses the GALPROP dataset [82, 83] and provide an alternative modeling of the radiation fields [84]. Figs. 9 and 10 show that, using this alternative ISRF model, the secondary fluxes and hence the total signals change by small amounts. Such a variation in the total signal in our energy range of interest is visible mostly for lower DM masses. With increasing values of the DM mass, this effect becomes less important. This statement is true for all representative annihilation channels considered in this work. Note from the discussion in sec. 2.1 that the ISRF densities enter into the signal calculation through the ICS emission power as well as through the energy loss of $e^{\pm}$.

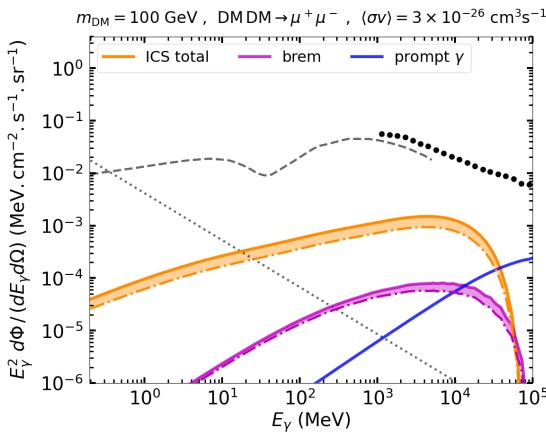

Figure 11: *DM-induced signals considering different **galactic magnetic field** models. The dashed-dotted curves (for the ICS and bremsstrahlung signals) correspond to the "MF3" model, that is comparatively stronger than the "MF1" model used for the solid curves here and in the main results. See [43] for a detailed discussion on these magnetic field models.*

### A.3 Magnetic field model

Fig. 11 shows the impact of considering two different Galactic magnetic field models, focusing again on the DM-induced secondary signals in the $\mu^+\mu^-$ annihilation channel for $m_{\mathrm{DM}} = 100$ GeV. The solid curves correspond to the model "MF1" (used in the main results) and are the same as the ones presented in fig. 1, while the dashed-dotted curves correspond to the model "MF3" which has a relatively larger strength in the magnetic field. See [43] for a detailed discussion on these models. The latter model causes a (modest) weakening of the DM secondary signal up to a factor of 2, which can be understood as the fact that the $e^\pm$ lose more energy to synchrotron radiation in the stronger magnetic field (see the discussion in sec. 2.1).

### A.4 Galactic gas maps

In Fig. 12 we show the level of changes in different secondary photon signals due to the variation of the Galactic gas densities. These are shown for $\mu^+\mu^-$ annihilation channel with $m_{\mathrm{DM}} = 10$ GeV (left panel), and $b\bar{b}$ annihilation channel with $m_{\mathrm{DM}} = 100$ GeV (right panel). For illustration, we vary the densities of all gas particles by a factor of 2 above and below their values in the base model (mentioned in Sec. 2.1.3 and used in our main results). The corresponding variations in different photon signals are indicated by the associated bands. The solid curves in each case correspond to the signals obtained using the base model. As expected, the main noticeable changes occur in the bremsstrahlung signal since its production relies on the density of the gas targets. There is also a very mild variation in the ICS signal through the energy loss term of $e^\pm$ that depends (through bremsstrahlung, Coulomb interaction and ionization processes) on the target gas densities. As can be seen from Fig. 12, the DM induced total photon signals for different cases remain almost robust against the above-mentioned changes in the gas density profiles.

### A.5 Atmospheric backgrounds

Among different future telescopes considered in this work, only E-ASTROGAM so far provides an estimate for the possible atmospheric background (see figure 18 of [18]). Such a background may arise due to various phenomena in the Earth's atmosphere including the atmospheric

lightnings and thunderstorms. In fig. 13 we show an example of the effects of the inclusion of this atmospheric background in the analysis. For the methodology to include such a background we follow [27]. As can be seen, for E-ASTROGAM the effects of the inclusion of the atmospheric background should be mild.

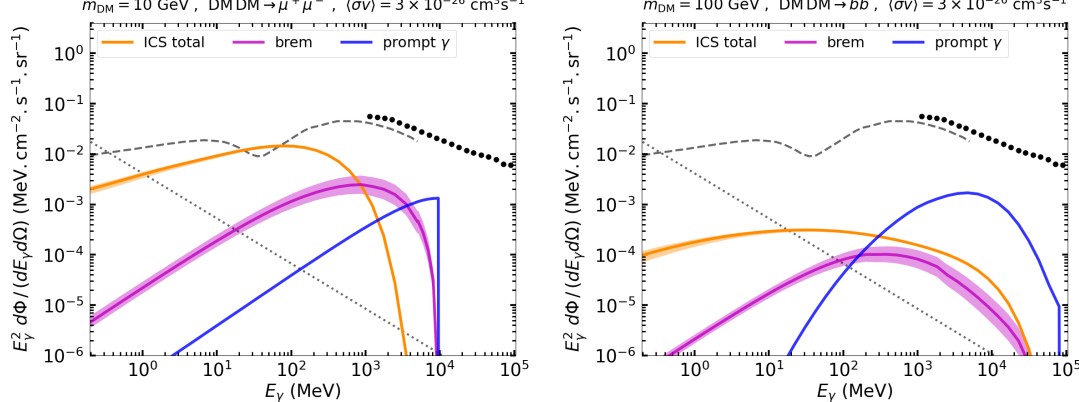

Figure 12: *DM-induced photon signals considering a factor of 2 **variation** (in both directions) **in the density maps of all Galactic gas particles**. The corresponding variations in different secondary photon signals are indicated by the associated bands. These are shown for $\mu^+\mu^-$ annihilation channel with $m_{\mathrm{DM}} = 10$ GeV (left panel), and $b\bar{b}$ annihilation channel with $m_{\mathrm{DM}} = 100$ GeV (right panel). The solid curves in each case correspond to the signals obtained following Sec. 2.1.*

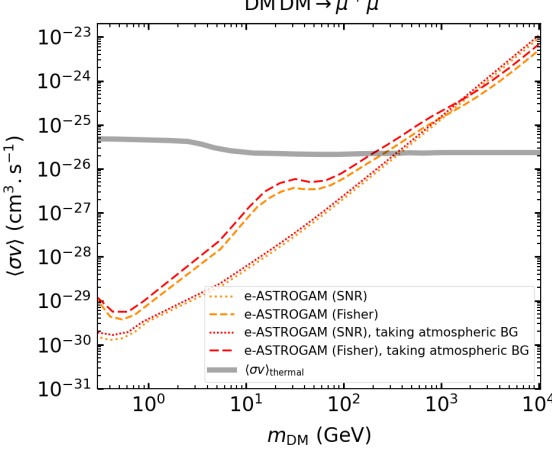

Figure 13: *Effects of the inclusion of the possible **atmospheric background** for E-ASTROGAM. The red curves are obtained by taking into account the atmospheric background estimated in [18], while the orange curves do not consider this and are the same as the ones shown in fig. 3.*

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
