# Peer review of "Prospects of future MeV telescopes in probing weak-scale Dark Matter"

_SciPost Physics, doi:SciPost Phys. 19, 080 (2025)_

## Round 2 · Referee Report · Anonymous (Referee 1) · 2025-5-15

Strengths

1.) The paper makes a very important point, which the community has yet to fully appreciate. MeV instruments can have world-leading sensitivity to GeV dark matter annihilation based on the secondary gamma-ray emission processes of electrons produced in the dark matter annihilation event.

2.) The results well-presented and thoroughly discussed. Uncertainties from the DM halo profile, galactic ISRF, and galactic magnetic field profile, are well considered and use world-leading models.

Weaknesses

1.) The actual sensitivity of MeV instruments is somewhat overstated, because real-world systematics (uncertainties in the astrophysical background model, and the effect of cosmic-ray diffusion) are not included. I believe that a similar simplified analysis written before the launch of the Fermi-LAT would easily indicate that the Fermi-LAT will rule out thermal dark matter up to the TeV range, a result that has not been realized because systematic uncertainties are dominant.

Report

I have carefully read through the paper "Prospects of future MeV telescopes in probing weak-scale Dark Matter" by Cirelli and Kar. I find the paper to be in general well-written and worth publishing after some modifications. I have two major scientific comment and a one more minor concern.

(1) My primary scientific concern regards the usage of either the SNR or Fisher information in order to set the constraints on the dark matter annihilation cross-section. I believe that both of these approaches focus on the size of the putative dark matter signal compared to the statistical fluctuations of the astrophysical background model.

However, we know that the systematic uncertainties on the astrophysical background are highly subdominant compared to the systematic uncertainties in the astrophysical background modeling -- which tends to significantly weaken the resulting dark matter constraints. For example, using the methods specified in the text, I assume that the constraints on a ~50 GeV dark matter particle annihilating to bb using Fermi-LAT data would be on the order of ~1e-28, however, we know from the Galactic Center excess that a dark matter particle with an annihilation cross-section two orders of magnitude larger remains consistent with the data.

I think the results of the paper would be strengthened if the systematic errors from different reasonable astrophysical background models would be integrated into the result. For example, one could take existing diffuse emission models (e.g., by comparing the residuals from two different astrophysical diffuse emission models, e.g., Models A/F/O which have been used in studies of the galactic center excess.) Alternatively, it would be worthwhile to show how the constraints react if 10% systematic errors were added into the diffuse modeling (which will cause things to approach closer to the conservative limits).

2.) My second major scientific concern regards the effect of cosmic-ray propagation on the strength of the constraints. In the paper, it is mentioned that cosmic-ray propagation should not affect these limits. However, the results in the appendix show that this result should approach an order of magnitude. It seems like this is really a well-motivated addition to the main text, and I would advocate using these models as the primary limits on the dark matter cross-section, rather than using the no-diffusion case, which we know is unlikely to represent the GC environment.

Additionally, it appears that DRAGON is being used in order to calculate the diffusion of cosmic-rays in the astrophysical background -- this makes the paper not self-consistent. The background is being generated assuming cosmic-ray propagation (which will decrease the leptonic signal near the GC) but the signal is being calculation without this dilution effect.

However, in the favor of the authors - I wouldn't put a convection velocity as large as 200 km/s as default, but I think it would make sense to show some version of a Min/Max model that is customized to the GC (choosing two different reasonable convection velocities and diffusion constants.). These can be self-consistently applied to both the signal and background (which is beneficial, because the high-convection velocities that will decrease the signal will also minimize the background).

3.) A more minor comment: Figures 3 and (especially) 5 are quite hard to read given the large number of lines in different colors and line-widths on them. I would try to sub-divide this into additional figures, so that the key results are easier for the reader to interpret.

Requested changes

1.) Include uncertainties in the astrophysical background model as a systematic uncertainty, which will weaken the sensitivity of MeV gamma-ray data to GeV-scale dark matter.

2.) Include propagation of cosmic-ray electrons for the dark matter induced signal. The discussion in the appendix is quite nice in defining an upper limit on the degree to which diffusion can affect the signal. However, a reasonable cosmic-ray diffusion model should be used as the default assumption for the analysis.

3.) Divide Figures 3 and 5 into more sub-figures, so that the results are easier to read.

Recommendation

Ask for major revision

  • validity: good
  • significance: high
  • originality: high
  • clarity: high
  • formatting: excellent
  • grammar: excellent

Author:  Arpan Kar  on 2025-07-29  [id 5688]

(in reply to Report 1 on 2025-05-15)

Dear Referee,

We would like to thank you for carefully going through the manuscript and providing us with your detailed and in-depth comments. Please find in the attached pdf ("reply-to-referee-1_incl-revised-draft.pdf") our responses in the first two pages. In the same pdf file we also attach (below our responses) the modified version of our paper highlighting the implemented modifications in `red' color, so that you can spot them easily.

Best regards,
the authors.

Attachment:

reply-to-referee-1_incl-revised-draft.pdf

Author:  Arpan Kar  on 2025-06-12  [id 5564]

(in reply to Report 1 on 2025-05-15)

Dear Referee,

We would like to thank you for carefully going through the manuscript and providing us with your detailed and in-depth comments. Please find our responses below in the attached pdf.
We will upload during the re-submission process a modified version of the manuscript that we have prepared highlighting the changes incorporated to address your comments.

Best regards,
the authors.

Attachment:

author-reply-referee1.pdf

---

## Round 2 · Referee Report · Anonymous (Referee 2) · 2025-7-3

Report

This paper explores the prospects of detecting weak-scale dark matter via secondary photon emission (inverse Compton scattering and bremsstrahlung) with future MeV gamma-ray telescopes such as AMEGO, E-ASTROGAM, and MAST. The authors use both semi-analytic and numerical methods to model the DM-induced signals and present sensitivity projections based on two complementary statistical approaches. While the idea of leveraging secondary emissions – in particular in the light of upcoming experiments – has already been introduced in the literature before, this work provides a timely and systematic assessment of its potential with upcoming instruments, which is likely to be of interest to the community.

However, before I can recommend the paper for publication, I believe the following points should be addressed:

1) The photon flux from DM-induced ICS and bremsstrahlung is computed using a semi-analytic approach developed in the authors’ earlier work ([22], [43]), based on fixed ISRF and gas models. This is a common and reasonable choice for a first estimate. The authors include a minimal cross-check in Appendix A2, showing that switching to a GALPROP-based ISRF model (“Porter2006”) results in only modest changes in the ICS flux – but this check is performed for a single final state (μ⁺μ⁻) and two DM masses (1 GeV and 100 GeV), without discussion of whether this behavior is representative of other channels or masses. Since the strength of the paper lies in its systematic assessment of future telescope sensitivity, the robustness of the secondary signal estimates is important. The paper would benefit from either a brief numerical extension of this cross-check or, at minimum, a clear justification of why the findings are expected to generalize. Similarly, the bremsstrahlung signal relies on fixed gas inputs from [43], with no cross-checks shown. Clarifying the potential variation – even qualitatively – would help the reader assess how much the projected sensitivity depends on these modelling assumptions.

2) The paper presents projected sensitivities using both a simplified SNR method and a Fisher matrix approach. Interestingly, the SNR method often leads to stronger sensitivity projections – a behavior that is not commented on in the text. Since the Fisher method is generally expected to provide a more realistic estimate by accounting for background uncertainties and spectral shape differences, it would be helpful if the authors briefly commented on the origin of this behavior. In particular, clarifying the circumstances under which the SNR method yields better projections (e.g. due to its more optimistic assumptions) and why the trend reverses for TeV-scale DM in leptonic channels would help readers interpret the results correctly.

3) While DRAGON2 and GALPROP implement the same physical processes, a brief comment in Appendix A4 on their compatibility would help readers who are not closely familiar with both codes understand why the full-propagation cross-check remains robust despite using a different tool than the one employed for background modelling.

Recommendation

Ask for major revision

  • validity: good
  • significance: good
  • originality: high
  • clarity: high
  • formatting: excellent
  • grammar: excellent

Author:  Arpan Kar  on 2025-07-29  [id 5689]

(in reply to Report 2 on 2025-07-03)

Dear Referee,

We would like to thank you for carefully going through the manuscript and providing us with your detailed and in-depth comments. Please find in the attached pdf ("reply-to-referee-2_incl-revised-draft.pdf") our responses in the first two pages. In the same pdf file we also attach (below our responses) the modified version of our paper highlighting the implemented modifications in `blue' color, so that you can spot them easily.

Best regards,
the authors.

Attachment:

reply-to-referee-2_incl-revised-draft.pdf

---

## Round 3 · Author Response

Please see the responses to the reports and the modified version of the draft.

---

## Round 3 · List of Changes

Please see the two attached pdf files, each one containing our responses to the corresponding referee and the modified draft with the changes marked in red (for referee 1) and blue (for referee 2) colors.

---

## Editorial Decision

published